# A Review of Botanical Extracts with Repellent and Insecticidal Activity and Their Suitability for Managing Mosquito-Borne Disease Risk in Mexico

**DOI:** 10.3390/pathogens13090737

**Published:** 2024-08-29

**Authors:** Josselin Carolina Corzo-Gómez, Josué Vidal Espinosa-Juárez, Jose Carlos Ovando-Zambrano, Alfredo Briones-Aranda, Abumalé Cruz-Salomón, Héctor Armando Esquinca-Avilés

**Affiliations:** 1Escuela de Ciencias Químicas, Universidad Autónoma de Chiapas, Ocozocoautla de Espinosa 29140, Chiapas, Mexico; josue.espinosa@unach.mx (J.V.E.-J.); carlos.ovando@unach.mx (J.C.O.-Z.); dr.abumale@gmail.com (A.C.-S.); 2Laboratorio de Farmacología, Facultad de Medicina Humana, Universidad Autónoma de Chiapas, Tuxtla Gutiérrez 29050, Chiapas, Mexico; alfred725@hotmail.com

**Keywords:** repellent, bioinsecticide, plants, vector control, arboviruses

## Abstract

Among the main arboviruses affecting public health in tropical regions are dengue, zika, and chikungunya, transmitted mainly by mosquitoes of the *Aedes* genus, especially *Aedes aegypti*. In recent years, outbreaks have posed major challenges to global health, highlighting the need for integrated and innovative strategies for their control and prevention. Prevention strategies include the elimination of vectors and avoiding mosquito bites; this can be achieved through the use of bioinsecticides and repellents based on plant phytochemicals, as they offer sustainable, ecological, and low-cost alternatives. Mexico has a variety of plants from which both extracts and essential oils have been obtained which have demonstrated significant efficacy in repelling and/or killing insect vectors. This review examines the current knowledge on plant species found in Mexico which are promising options concerning synthetic compounds in terms of their repellent and insecticidal properties against mosquitoes of the genus *Aedes* and that are friendly to the environment and health.

## 1. Introduction

Arboviruses are viruses transmitted to vertebrate hosts by arthropods (e.g., ticks and mosquitoes), causing diseases that range from asymptomatic to life-threatening. Some of the most recognized arboviruses impacting public health systems are the dengue virus (DENV), chikungunya virus (CHIKV), and zika virus (ZIKV). These viruses are transmitted by mosquito vectors, particularly of the *Aedes* genus, in the American continent; the main vector is *Ae. aegypti* (Linnaeus, 1762) [1]. 

Over the past decade, the circulation pattern of arboviruses in the American continent has shown various fluctuations, largely due to the different epidemiological scenarios created by DENV compared to the introduction of CHIKV in 2013–2014 and ZIKV in 2015. Despite these changes, DENV circulation has remained predominant. In 2021, there were 1,268,767 cases of dengue, 138,358 cases of chikungunya, and 23,142 cases of zika reported in the Americas [2]. Based on the geographical distribution of arbovirus cases, the highest number of dengue cases was reported in Brazil with 975,474 cases (76.9%), followed by Colombia with 53,334 cases (4.2%), Peru with 49,274 cases (3.9%), Nicaragua with 36,741 cases (2.9%), and Mexico with 36,742 cases (2.9%). Notably, Mexico exhibits the simultaneous circulation of all four DENV serotypes, with nearly 80% of dengue cases occurring in the southeastern region, including the states of Veracruz, Yucatán, Guerrero, Chiapas, Oaxaca, and Tabasco. Since the first case of the autochthonous transmission of CHIKV in October 2014, 12,588 cases have been reported, indicating active transmission and its dissemination throughout the country [3]. For zika, the first case in Mexico was reported in November 2015, in a patient without travel history to areas where the virus was circulating. From this first case until December 2016 a total of 7427 cases were reported [4]. In February 2017, the first case of microcephaly associated with zika was reported [5]. These outbreaks have become increasingly frequent and now represent significant challenges to global health. Consequently, the World Health Organization (WHO) has recognized the need for integrated and innovative strategies for the control and prevention of arboviral diseases, primarily through the use of chemical compounds to combat the mosquito vector.

However, due to the detrimental effects of synthetic products used for decades to reduce the transmission of mosquito-borne diseases, coupled with current resistance problems that have made it difficult to eliminate the vector, there is a growing interest in exploring plant-derived products to control mosquito populations and prevent their bite. Therefore, it is crucial to develop control strategies that are safer, more effective, and more sustainable compared to traditional chemical interventions. 

This review presents updated information on plant-based compounds characterized by their repellent and/or insecticidal action, suggesting them as new possibilities for managing mosquito populations in a biodegradable, sustainable, and environmentally friendly manner. Furthermore, given the great biodiversity of plants in Mexico, the primary plants cultivated in the country that have been identified as exhibiting bioactive compounds with repellent and/or insecticidal properties are presented and characterized, representing a substantial opportunity for developing natural products to combat vector-borne diseases. 

### 1.1. Transmission of Arboviruses: Dengue, Zika and Chikungunya

Arboviruses such as dengue, zika, and chikungunya have transmission cycles involving vertebrates (birds or mammals) and hematophagous vectors [6]. These viruses follow two infectious cycles: the sylvatic cycle, where viral replication occurs in animal reservoirs, and the urban cycle, specific to certain arboviruses, which facilitates human transmission (Figure 1) and epidemic outbreaks [7]. The urban cycle emerged when mosquitoes of the genus *Aedes* adapted their reproductive habits to the human environment, completing their life cycle in water stored in containers and transmitting the virus through bites from infected females. In contrast, the sylvatic cycle involves non-human primates as natural reservoirs, with transmission by several species of *Aedes* [8,9].

To transmit the infection, female mosquitoes harbor the virus in their midgut, which then disseminates to the salivary glands, where it replicates and is transmitted to another host during a bite [10]. The extrinsic incubation period of arboviruses varies from 7 to 14 days, depending on the virus and the viral load. Infection in *Ae. aegypti* is permanent, allowing the mosquito to transmit the virus throughout its life and occasionally vertically to its offspring, although vertical transmission is minimal, with an infection rate of approximately 1 per 1000 eggs [11]. Mosquito proliferation increases during the rainy season, heightening the risk of arbovirus transmission in summer [12]. 

The most common mode of transmission of zika, dengue, and chikungunya is through the bite of their common vector, generating an urban, human–mosquito–human transmission cycle. However, the vertical transmission of dengue and chikungunya during pregnancy has also been reported [13]. For ZIKV, *Ae. aegypti* is the primary vector responsible for most outbreaks and the main mode of transmission. Additionally, there is evidence that ZIKV can be transmitted directly, from one human to another through other routes. For example, there have been cases of sexual transmission among couples with a history of travel to endemic areas [14], as well as cases of transmission through blood transfusions and vertical transmission from mother to fetus [15]. This last route has been a significant concern due to the complications in neuronal development that the virus can cause in the fetus and newborn during any trimester of pregnancy, leading to microcephaly or other neurological diseases [16].

### 1.2. Origin and Expansion of Mosquito Vectors

The genus *Aedes* comprises over 950 species with a global distribution, and these are significantly influenced by human activities such as travel, trade, urban development, and climate change. These mosquitoes belong to the class insecta, order diptera, family culicidae, and subfamily culicinae [12]. 

*Aedes aegypti*, originally from Africa, and *Ae. albopictus*, from Asia, have become significantly globalized. *Aedes aegypti* is believed to have been introduced to America in barrels of water on ships during European colonization and the slave trade [17]. Their adaptation to synanthropic environments has facilitated their proliferation, exacerbated by urbanization and increased international travel and trade, making them effective vectors of arboviruses [18,19]. Additionally, climate change has contributed to their expansion, allowing these mosquitoes to colonize new geographic areas [20]. 

#### Behavior of Mosquito Vectors

Historically, *Ae. aegypti* has been an efficient vector for various arboviruses due to its preference for human blood and its behavior of biting multiple hosts in a single feeding [21]. In contrast, *Ae. albopictus* has a diurnal biting activity, feeds on a wider variety of mammals, and prefers colder habitats, making it less efficient as a vector. Furthermore, *Ae. aegypti* prefers indoor habitats, reproducing in water containers close to human dwellings. Its life cycle lasts from 8 to 15 days, with an adult phase of four to six weeks. This species is most active at dawn and dusk but can bite at different times of the day depending on the availability of blood sources [22].

### 1.3. Strategies for the Prevention and Control of Mosquitoes That Transmit Arboviruses

The prevention and control of *Ae. aegypti* are crucial for public health, especially in tropical and subtropical regions. Various control methods, including chemical, biological, and environmental management, target different stages of the mosquito’s life cycle and are often applied simultaneously. The Mexican Official Standard (NOM-032-SSA2-2014) for the epidemiological surveillance, prevention, and control of vector-borne diseases (VBD) coincides with the guidelines and recommendations issued by the WHO/PAHO, carrying out an integrated management of these diseases and considering a series of control measures, both physical, chemical, and biological strategies, that are applied sequentially and/or synchronized to effectively reduce vector populations. The guidelines for physical control recommend the application of the safe housing and water strategy. Chemical control is carried out using insecticides aimed at different stages of the vector’s life cycle, from its immature to its adult stages, applying chemical, microbial, botanical, or miscellaneous larvicides or adulticides. The applied adulticides may be short-acting or residual, including pavilions impregnated with insecticide. The integrated management measures are applied with periodic application of these control measures throughout the municipality of the country. In addition, in situations with probable cases, targeted control is carried out in homes in areas of high entomological risk. As in many dengue-endemic countries, in Mexico, vector control for zika and chikungunya is also carried out as indicated for dengue, since it is the same species of mosquito that transmits it. Additionally, interest in surveillance, prevention, and control programs for VBD should be promoted among national research institutions—such as universities and higher education institutions that carry out basic and applied (operational) research, which serves to optimize and make sustainable the use of resources in the design of evidence-based strategies—in its epidemiological and entomological aspects, as well as in operational, administrative, and socioeconomic aspects, with particular emphasis on risk factors, for the implementation of methods that are more cost-efficient in control and for their evaluation, with the aim of reducing or interrupting these diseases [23]. 

Chemical control is a key strategy for combating arbovirus disease as it addresses both immature stages (eggs, larvae, pupae) and the adult mosquitoes. Personal protection through the use of domestic and commercial insecticides, as well as synthetic repellents, can also have a significant entomological and epidemiological impact [24]. However, the indiscriminate use of chemicals has led to numerous environmental issues, such as the pollution of water, air, and soil, and the development of insect resistance. For instance, 1,1,1-trichloro-2,2-bis(4-chlorophenyl)-ethane (DDT) had been extensively used for malaria mosquito control since 1939 and became the insecticide par excellence of that time [25,26]. Consequently, there is a need to enhance vector control methods to address these challenges and reduce the harmful impacts on ecosystems and human health. Plant-derived insecticides have emerged as a viable and environmentally sustainable alternative to synthetic compounds. These natural options are cost-effective and easily biodegradable, and their mode of action involves multiple mechanisms that precisely target pests [27]. Additionally, avoiding mosquito bites is a low-cost and essential strategy for preventing infection and transmission. Mechanical barriers such as mosquito nets can prevent contact between people and mosquitoes, offering an economical, environmentally friendly, and health-safe option. However, their effectiveness depends on the quality of the net material, correct installation, and proper use.

The use of insect repellents is also being considered; currently, laboratory-synthesized chemical repellents such as N,N-diethyl-toluamide (DEET), ethyl butylacetylaminopropionate (IR3535), and picaridin are effective and widely used. DEET, the most common repellent, interferes with the olfactory and gustatory systems of mosquitoes and provides up to 8 h of protection at concentrations above 40–45%. However, its effectiveness can vary among mosquito populations, and there are concerns about toxicity and irritation in humans [28]. In contrast, IR3535 is known for its low toxicity and high skin tolerance, and is used in various formulations to prolong its repellent effect, offering protection of a similar or slightly shorter duration than that of DEET, although at varied concentrations, ranging from 5 to 25% [28,29]. Picaridin, a promising alternative to DEET, offers comparable protection with less skin irritation, demonstrating a 95% repellent efficacy for 8 h at concentrations of 20%. Microencapsulation techniques have been developed to extend its protection time, improving user compliance [30]. 

## 2. Plants as a Potential Source for the Prevention and Control of Mosquito Bites

Controlling arbovirus vectors is a constant challenge. The use of plants can offer sustainable solutions to combat the spread of mosquito-borne diseases. Nowadays, compounds of plant origin have been discovered to exhibit insecticidal and repellent properties. Research on the insecticidal and repellent effects of plant-based compounds has shown promising results, with both extracts and essential oils from various plants, proving to be safe and effective. It has been reported that chemical compounds with mosquito-cidal activity are mostly secondary metabolites such as essential oils, alkaloids, phenols, terpenoids, steroids, and phenolics from different plants [31,32]. The ovicidal and larvicidal effects of plant extracts have been widely studied in the immobile stages of mosquitoes, i.e., the stages before they emerge as adults. Essential oils extracted from plants are rich sources of compounds with repellent activity, primarily derived from their various active constituents. These essential oils contain numerous bioactive compounds that have been extensively studied for their ability to repel mosquitoes or function as insecticides. Researchers have identified essential oil components such as terpenes, phenols, and aromatic compounds that demonstrate effective mosquito repellency and insecticidal properties.

Therefore, it is important to consider the properties of some plants that have been recognized for centuries, the volatile compounds of which, such as monoterpenes and sesquiterpenes, can be effective against mosquitoes [33]. Plant-based insect repellents have been used since ancient times, and their production and marketing are on the rise as plant extracts have emerged as an eco-friendly strategy for managing the transmission of arboviruses. Plants with repellent effects not only effectively combat the bite of the vector but also minimize environmental damage, being compatible with beneficial insects and wildlife. Recent studies have indicated that plant essential oils possess confirmed repellent qualities against *Aedes* mosquitoes. These essential oils are abundant in bioactive compounds such as citronella, limonene, and thymol, which have been shown to influence the sensory mechanisms of mosquitoes, thereby decreasing their ability to search for hosts. These findings support the idea that these bioactive compounds can play a pivotal role in altering the ability of mosquitoes to locate and feed on hosts [34,35]. 

On the other hand, insecticides of plant origin consist of several chemical compounds that act together in physiological and behavioral processes, unlike conventional insecticides that contain single-component bioactive compounds. This multi-component nature allows bioinsecticides to employ multiple mechanisms of action, which can help mitigate the development of resistance [36]. This highlights the versatility of plant-based insecticides, improving the efficacy and sustainability of insect pest control strategies. Consequently, plant-based insecticides offer a more environmentally friendly approach to controlling *Ae. aegypti* populations.

In line with the above, Mexico’s abundant plant biodiversity offers a significant opportunity for developing natural repellents and insecticides to combat vector-borne diseases. The country’s diverse flora include numerous plant species that have been traditionally used in medicine for their insect-fighting properties.

### 2.1. Citronella (Cymbopogoncus nardus)

The genus *Cymbopogon* includes 144 species native to Asia, Africa, Australia, and tropical islands [37]. It is known by various names such as lemongrass, barbed wire grass, silky heads, cochin grass, malabar grass, oily heads, citronella grass, and fever grass. It is distributed across over 40 countries, including Mexico. Particularly, *C. nardus*, commonly known as citronella grass, is a perennial aromatic plant from the Poaceae family. The ethnopharmacological properties of *Cymbopogon* species are well documented, with a broad range of uses in skin care products, fragrances, medicines (including antiprotozoal, antibacterial, anti-inflammatory, and anticancer medicines), and exhibit mosquito repellency and insecticidal properties due to their high concentration of volatile essential oils [38,39]. The therapeutic potential of these herbs is attributed to their content of phytochemicals, including hydrocarbons, alcohols, ketones, esters, phenols, flavonoids, tannins, terpenoids, carotenoids, and other volatile and non-volatile compounds. Essential oils, flavonoids, terpenoids, phenols, and tannins are key phytoconstituents that contribute significantly to the plant’s therapeutic properties. The essential oil of *Cymbopogon* species primarily contains monoterpenes, monoterpenoids, sesquiterpenes, sesquiterpenoids, and fatty alcohols like 1-octanol and 4-nonanol [40,41]. Members of the *Cymbopogon* genus, especially *C. citratus*, are commonly used as insect repellents against mosquitoes, houseflies, and fleas in various countries [42]. Essential oils from *C. citratus*, *C. nardus*, *C. martini*, and *C. flexuosus* have demonstrated effectiveness against mosquitoes such as *An. culicifacies*, *Cx. Quinquefasciatus*, and *Ae. aegypti* [43]. Additionally, *C. nardus* shows both adulticidal and larvicidal properties against *Ae. aegypti*, presenting a potent natural alternative to synthetic insecticides [44,45]. However, it is important to considerer that these affects may also impact non-target beneficial insects [46]. Despite this, *C. nardus* essential oil remains a promising biopesticide due to its multifunctional properties, including repellent activity, making it valuable for integrated pest-management. Given its global distribution, further research is needed to enhance understanding of its repellent and insecticidal properties, with the aim of developing natural products that can replace more aggressive chemical products, which often have negative side effects on health. 

### 2.2. Pepper (Piper aduncum)

*Piper aduncum*, a member of the Piperaceae family, is a tree native to the coasts and jungles of the Americas, growing up to 3–4 m in height. Commonly known as matico, achotlín, or cordoncillo, it features a woody, branched stem that is green or pale gray with light green leaves. The plant produces tiny fruits containing small black seeds which aid in its reproductive cycle and dispersal. In traditional medicine, *P. aduncum* is valued for its potential effects, including anti-inflammatory, expectorant, antitussive, healing, and antiseptic properties [47]. 

Recent studies have explored the use of essential oils from the leaves and stems of *P. aduncum* as an alternative to synthetic pesticides, particularly to address insect resistance to commercial pesticides [48,49]. The essential oils of this plant contain two major groups of compounds: phenylpropanoids and monoterpenes. Notable molecules in its makeup include γ-terpinene, p-cymene, limonene, β-asarone, (E)-anethole, (E)-β-caryophyllene, γ-terpinene, p-cymene, α-pinene and β-pinene, dillapiol, (E)-anethole, β-asarone, 1,8-cineole, (E)-nerolidol, dillapiole, and asaricin [50]. This chemical profile is significant for identifying compounds useful in health products. Essential oils from *P. aduncum* exhibit promising insecticidal activity against *Ae. aegypti*, with research showing up to 90% larvicidal activity at 100 ppm, effective against both pyrethroid-resistant and -susceptible strains [50]. Another study on the essential oils of different species of *Piper* (*P. betle*) also demonstrated larvicidal and adulticidal effects on *Ae. aegypti,* with larvicidal activity observed at various concentrations and 100% adult mortality at 2.5 μL/mL within 15–30 min, highlighting its potential as a bioinsecticide [51].

Regarding repellent effects, *P. aduncum* essential oil achieved a 93.5% repellent rate with 180 min of protection at a concentration of 1000 mg/mL, showing its natural efficacy against *Ae. aegypti* adults [52]. Additionally, the oil demonstrated an immediate reduction in mosquito bites on human skin [49]. When tested against *Ae. albopictus*, the essential oil provided 83.3% protection after 4 h, 64.5% after 6 h, and 51.6% after 8 h, indicating moderate protection [53].

### 2.3. Black Pepper (Piper nigrum)

Black pepper, scientifically known as *P. nigrum*, belongs to the Piperaceae family and is primarily cultivated for its fruit, drupe, which measures around 5 mm in size. The fruit is dried and used as a condiment, either whole or ground into a fine powder, making it a staple ingredient in culinary practices worldwide. Introduced to Mexico, *P. nigrum* is a climbing plant that can grow over 4 m in height with support from trees, trellises, or other structures. The plant spreads efficiently by rooting from its stems upon contact with the ground, facilitating rapid colonization. Its leaves, arranged alternately along the stems, are undivided and typically measure 5–10 cm in length and 3–6 cm in width. The small flowers of *P. nigrum* form drooping clusters in the leaf axils, measuring 4–8 cm long, and expand to approximately 7–15 cm as they mature into fruits [54]. 

Currently, the key phytoconstituents that have been identified in the essential oil of black pepper include (E)-caryophyllene, limonene, sabinene, α-terpineol, borneol, terpinen-4-ol, β-caryophyllene, α-pinene, β-pinene, 3-carene, eugenol, and methyl eugenol [55]. The main bioactive compound of black pepper, piperine, has been extensively studied for its anti-inflammatory and antioxidant properties, contributing to its effectiveness in fighting premature aging. Additionally, piperine has been reported to exhibit larvicidal properties. Research by Samuel et al. [56] investigated the larvicidal effects of ground black pepper and piperine against third- and fourth-instar *Anopheles* larvae from insecticide-resistant and susceptible strains. They found that both black pepper and piperine caused high mortality rates in these larvae, with black pepper showing significantly higher toxicity compared to piperine. Similar larvicidal effects were observed with essential oils on *An. gambiae* larvae, achieving a 100% mortality with an LC_50_ value of 15 ppm for *P. nigrum* [56]. Furthermore, the essential oil of *P. nigrum* has demonstrated insect repellent properties. Amer and Mehlhorn found that the essential oil provided a relatively short protection time, with 64.9% repellency for *Ae. aegypti* at 90 min, 61.9% for *An. stephensi* at 180 min, and 100% for *Cx. quinquefasciatus* at 480 min. These findings align with those reported by Tawatsin et al. [57], which noted a modest protective effect against *Ae. aegypti* bites.

### 2.4. Garlic (Allium sativum)

*Allium sativum*, commonly known as garlic, belongs to the Amaryllidaceae family and is highly valued both as a culinary ingredient and in traditional medicine for various ailments. Although it originated in Asia, garlic cultivation has spread to regions such as Greece, Rome, India, Egypt, Europe, China, and Mexico [58]. *Allium sativum* is rich in sulfur-containing phytochemicals, including allicin, diallyl disulfide (DADS), vinyldithiins, ajoenes (E-ajoene, Z-ajoene), and diallyl trisulfide (DATS), as well as essential micronutrients like selenium (Se) [59,60]. These organosulfur compounds contribute to its potent medicinal properties. The chemical composition of garlic essential oil includes dimethyl trisulfide, diallyl disulfide, diallyl sulfide, diallyl tetrasulfide, and 3-vinyl-[4H]-1,2-dithiine [61]. Research has demonstrated a broad range of pharmacological properties for *A. sativum*, including antioxidant [62], hypoglycemic [63], anti-inflammatory [64], anticancer [65], antimicrobial [66], and hepatoprotective [67] activities. Garlic has also been suggested as a natural insect repellent. Stjernberg and Berglund [68] noted that garlic consumption might reduce tick bites in navy recruits, indicating its potential as a natural repellent. Comparative studies of garlic extracts found that aqueous extracts had a lower repellent efficacy against adult *Ae. aegypti* and *An. stephensi* than DEET, while oil extracts showed significant repellent effects, with 20% oil providing mean protection times of 130 min against *An. stephensi* and 140 min against *Ae. aegypti* [69]. Furthermore, research by Prada-Ardila et al. [70] suggested that garlic extracts have larvicidal effects on *Ae. aegypti* stage IV larvae, achieving an 85% mortality at 2000 ppm after 48 h. Additionally, garlic oil exhibited toxicity against mosquito larvae, with LC_50_ values of 0.314 ppm and 0.40 ppm for *Ae. aegypti* and *An. stephensi*, respectively, after 24 h, and LC_90_ values of 12,323 ppm and 2.13 ppm for the same species [69].

### 2.5. Lemongrass (Cymbopogon citratus)

*Cymbopogon*, a genus of around 55 species, is predominantly found in tropical and subtropical regions of Asia but is also successfully cultivated in Africa, South and Central America, and other tropical countries, including Mexico. This widespread presence highlights the genus’s adaptability to various climates and environments, making it a valuable resource for traditional medicine, culinary practices, and the aromatic industry [71]. Characterized by its herbaceous nature, *Cymbopogon* typically grows in dense clusters, reaching heights of approximately 1.8 m and widths of around 1.2 m, with a short rhizome [41]. *Cymbopogon citratus*, commonly known as lemongrass, is notable for its leaves and whole-plant usage. The essential oil of *C. citratus* contains various compounds such as citral α, citral β, nerol, geraniol, citronellal, terpinolene, geranyl acetate, myrcene, terpinol methylheptenone, limonene, and geranial. Additionally, it includes phytoconstituents like flavonoids and phenolic compounds, including luteolin, isoorientin 2′-O-rhamnoside, quercetin, kaempferol, and apigenin [72,73]. 

Several studies have demonstrated a range of pharmacological actions for *C. citratus*, including antiamoebic [74], antibacterial [75,76], antidiarrheal [77], antifilariae [78], antifungal [79,80], and anti-inflammatory properties. Its repellent effect has been shown against sandflies, vectors of leishmaniasis, as reported by Kimutai et al. in 2017 [81]. The study found that the repellent effect increased with higher doses of *C. citratus* essential oil, with a 50% effective dose of 0.04 mg/mL, a 90% effective dose of 0.79 mg/mL, and a 100% repellency at 1 mg/mL; a lower dose of 0.5 mg/mL achieved 89.13% repellency. Additionally, its repellent effect has been demonstrated against other insects, such as the peanut beetle (*Ulomoides dermestoides* by Fairmaire, 1893), a global pest of stored grains [82]. The insecticidal properties of *C. citratus* essential oil have been noted, with effective larvicidal activity against third and fourth stage *Ae. aegypti* larvae, showing an LC_50_ of 123.3 ppm [83]. This finding complements previous reports by Cavalcanti et al. who recorded an LC_50_ of 69 ppm for the same mosquito species [84].

### 2.6. Cinnamon (Cinnamomum zeylanicum)

*Cinnamomum zeylanicum*, a tree native to Asia and classified within the Lauraceae family, grows to a height of 10–15 m. It features grayish-brown bark, woody stems, and vibrant green, oval, pointed leaves. The tree produces hermaphroditic flowers in white or greenish-yellow and long, ellipsoidal bluish-black berries. It thrives in warm, humid climates and is cultivated from Southeast Asia to South America and Mexico [85]. 

The plant’s chemical composition varies across its parts, with cinnamaldehyde predominant in the bark, eugenol in the leaves, and camphor in the roots [86]. Essential oils from the bark, leaves, and root bark show notable chemical differences, reflecting their distinct pharmacological properties [87]. Essential oils from the bark, for instance, contain significant amounts of trans-cinnamaldehyde, eugenol, and linalool, comprising 82.5% of the total composition [88]. The concentration of trans-cinnamaldehyde in the bark oil ranges from 49.9% to 62.8% [89,90]. Both cinnamaldehyde and eugenol are key components of the plant, underscoring their importance in its chemical profile [91].

Research highlights the diverse health benefits of *C. zeylanicum*, including anti-inflammatory, antimicrobial, cardiovascular, cognitive, and cancer-preventive effects [92]. Additionally, *C. zeylanicum* exhibits repellent properties, with studies showing 100% repellency for up to 8 h against *Anopheles* mosquitoes [93]. Its essential oils, particularly from the leaves, have shown significant larvicidal potential against *Ae. aegypti* larvae, outperforming those from the flowers [94,95]. Furthermore, metabolites like cinnamic acid have also demonstrated larvicidal effects on *Ae. aegypti* [96].

### 2.7. Neem (Azadirachta indica)

*Azadirachta indica*, commonly known as neem or Indian nimbus, is a tree native to India and Burma, and is widely cultivated in tropical regions, including Mexico. It typically reaches heights of 15–20 m and maintains its evergreen foliage throughout most growing seasons, shedding leaves only under extreme conditions. Neem trees flower between April and May, with fruiting occurring from May to August [97]. 

Neem is renowned for its diverse biological and pharmacological activities, including antibacterial, antifungal, and anti-inflammatory effects [98,99]. The primary product derived from neem is its oil, extracted from the seeds, which contains several active compounds such as azadirachtin, meliantriol, salannin, deacetyl salannin, nimbin, deacetyl nimbin, nimbide, and nimbolides [100]. Azadirachtin, a limonoid, is particularly notable for its pest-repellent and insecticidal properties. It mimics insect hormones (ecdysone) and disrupts the normal process of insect metamorphosis by acting as an antagonist to ecdysone. This interference affects the levels of ecdysteroids and juvenile hormones in insects, leading to growth abnormalities and developmental issues [101]. 

*Azadirachta indica* oil’s repellent efficacy has been demonstrated against various mosquito species, particularly those in the *Anopheles* genus. In field trials, a 20% neem oil solution showed an average repellency of 71%, providing up to 3 h of protection against *An. arabiensis* [102]. These findings underscore the potential of neem as a natural vector control agent. However, due to the behavioral differences among mosquito species, further research is needed to optimize neem’s effectiveness across various mosquito types.

### 2.8. Basil (Ocimum basilicum)

*Ocimum basilicum*, a member of the Lamiaceae family, originates from India and is extensively cultivated in tropical regions, including Mexico. Known for its medicinal properties and distinctive flavor, basil grows as a perennial in tropical climates, reaching heights of 30–130 cm. It features bright green, oval-shaped leaves with serrated edges and a silky texture. A notable characteristic of *O. basilicum* is its terminal flower spikes, which bear tubular flowers in white or purple hues. The plant is typically propagated through seeds or cuttings and thrives in sunny locations, although some shade may be beneficial in regions with very hot summers. Optimal growing conditions include fertile, well-drained soil that is kept consistently moist [103,104,105].

The essential oil of basil is rich in bioactive phytochemical compounds, including alkaloids, phenols, flavonoids, tannins, saponins, reducing sugars, cardiac glycosides, steroids, and glucosides. Key molecules identified in basil’s essential oil include linalool, thymol, cineole, eugenol, methylchavicol, 2-(2-butoxyethoxy)ethanol, cedrelanol, methyleugenol, 2,4-di-tert-butylphenol, and phytol [106].

*Ocimum basilicum* exhibits a range of pharmacological effects, including anti-inflammatory, antifungal, antibacterial, antioxidant, healing, and antiviral properties [106]. Although its repellent activity is less well-documented, a study assessed the effectiveness of topical applications of *O. basilicum* essential oil at three concentrations (0.02, 0.10, and 0.21 mg/cm^2^) on volunteers’ forearms. The study demonstrated high repellency against *Ae. aegypti* with an ED_50_ of less than 0.045 mg/cm^2^, though the protection time was relatively short, approximately 40 min [107].

Further evaluations of basil’s repellent effects have involved methanol, acetone, and petroleum ether extracts. The methanolic extract showed the lowest repellent activity at 77.4% with a dose of 6.7 mg/cm^2^, while the petroleum ether extract achieved 98.1% repellency at the same dose. Additionally, the methanolic extract was found to be effective against adult mosquitoes [108]. In contrast, Lopes et al. [109] reported varying larvicidal effects of *O. basilicum* essential oil against third-stage *Ae. aegypti* larvae, with concentrations of 0.05–0.25 µL/mL causing from 73.2% to 64% larval mortality. Ethanolic and hexane extracts of basil leaves also demonstrated high larvicidal efficacy, with mortality rates of up to 100% [106,110].

### 2.9. Guava (Psidium guajava)

*Psidium guajava*, better known as guava, is a shrub that is native to the American continent and widely cultivated in tropical and subtropical regions, with Mexico being one of the world’s leading producers [111,112,113]. This shrub or small tree can reach heights from 7 to 10 m and has a spread of up to 25 cm in diameter. It features smooth, thin, copper-brown bark, green, evergreen leaves, and distinctive white flowers with 4–5 petals and numerous stamens. The fruit is a juicy berry, approximately 5 cm in diameter, and varies in color from yellow to deep pink, with a slightly acidic taste. Guavas can be round, ovoid, or pear-shaped [111,112]. 

The chemical composition of guava includes tannins, phenols, flavonoids, saponins, carbohydrates, alkaloids, sterols, and terpenoids [113]. The leaves are particularly rich in phenolic compounds, flavonoids, carotenoids, terpenoids, triterpenes, and essential vitamins and minerals such as calcium, potassium, sodium, magnesium, iron, sulfur, and vitamins B and C [114]. Due to its widely medicinal use, the most analyzed part of the plant is its leaves, where the presence of phenolic compounds [115], flavonoids, carotenoids, terpenoids, and triterpenes has been reported [116]; as well as vitamins and minerals, such as calcium, potassium, sodium, magnesium, iron, sulfur, and vitamins B and C [114]. Notably, β-caryophyllene, a compound found in guava leaf extracts and essential oils, has gained attention for its potential anti-inflammatory [117] and antimicrobial properties [118], although its exact mechanisms are not yet fully understood. 

Guava leaves also exhibit larvicidal activity. Rwang et al. found that high concentrations of guava aqueous extracts (15%, 20%, and 25%) were effective against mosquito larvae within 20 min, while guava ethanolic extracts demonstrated larvicidal effects at all concentrations, suggesting potential for sustainable malaria vector control [119]. Similarly, Harun et al. (2024) reported that all guava solvent extracts showed larvicidal activity, with the methanolic extract exhibiting the highest mortality against *Ae. aegypti* larvae [120]. However, research on the repellent properties of guava leaf extracts or essential oils is limited, making it a promising area for future investigation, especially given its widespread availability.

### 2.10. Peppermint (Mentha piperita)

*Mentha piperita*, commonly known as peppermint, is a perennial herbaceous plant from the Lamiaceae family. It features highly branched stems that grow between 30 and 70 cm tall and originate from an extensive underground rhizome. Peppermint leaves are petiolate, opposite, and oval, measuring from 4 to 9 cm in length and from 2 to 4 cm in width, with a sharp apex and serrated edges. The upper surface of the leaves is dark green with fine red veins arranged in a pinnate pattern [121]. Originally native to Europe and Asia, peppermint is now cultivated in various regions worldwide, including Africa, Australia, North America, South America, and Mexico. 

The leaves of *M. piperita* are utilized for various purposes, including gastronomy, personal care products, and aromatherapy. In traditional medicine, peppermint has been employed to alleviate gastrointestinal issues. More recently, its analgesic, anti-inflammatory, antimicrobial, anticancer, and neuroprotective properties have been recognized [122,123,124]. These medicinal benefits are attributed to the presence of compounds such as phenols, alkaloids, tannins, saponins, flavonoids, carbohydrates, proteins, monoterpenes, terpenoids, and steroids, identified in both ethanolic and aqueous peppermint leaf extracts [125]. Additionally, peppermint essential oil contains phytochemicals like menthol, menthone, neomenthol, iso-menthone, menthofuran, and pulegone [124,125], the essential oil has shown repellent effects, as topical application in mice resulted in a significant reduction in mosquito bites. This repellent action was further demonstrated, using aqueous and ethanolic extracts, against *Ae. aegypti* and *An. stephensi*, with repulsiveness lasting up to 120 min [126,127]. Furthermore, the insecticidal properties of the plant were observed, with ethanolic and aqueous peppermint extracts resulting in 93% larval mortality against *Ae. aegypti* and *Cx. quinquefasciatus* [123]. 

### 2.11. Chamomile (Anthemis nobilis)

*Anthemis nobilis*, commonly known as chamomile, is a herbaceous plant from the Asteraceae family. Native to Western and Central Europe, it is now cultivated in various regions, including Australia, North America, and Mexico. In traditional medicine and the pharmaceutical and cosmetic industries, chamomile is valued for its essential oils. The plant’s flowers are often used to prepare infusions that address digestive issues and calm the nervous system [128]. *Anthemis nobilis* exhibits a range of medicinal properties, including anti-inflammatory, analgesic, sedative, antispasmodic, antimicrobial, and antioxidant effects. These benefits are attributed to compounds such as flavonoids, coumarins, volatile oils, terpenes, sterols, organic acids, and polysaccharides which are found in its aqueous extracts, as well as essential oil components like α-bisabolol, α-pinene, β-pinene, chamazulene, en-yne-dicycloether, farnesene, spiroether, and various esters [129].

Interestingly, *A. nobilis* has demonstrated both repellent and insecticidal properties. The essential oil of this plant has shown a significant repellent effect against *Ae. aegypti* and *Ae. albopictus* mosquitoes. In experiments, the oil applied to the hairless abdominal area of rats recorded effective repellency for up to 30 min post-exposure compared to a control area [130]. In addition to its repellent properties, the essential oil has also been evaluated for its insecticidal action. Bioassays involving female *Ae. aegypti* mosquitoes and the Caribbean fruit fly (*Anastrepha suspensa*) revealed that the essential oil effectively repelled *Ae. aegypti* and exhibited insecticidal activity against *A. suspensa* when the insects were exposed to feeding solutions containing various concentrations of the oil [131].

### 2.12. Jamaican Rosewood (Amyris balsamifera)

*Amyris balsamifera* is a species of tree belonging to the Rutaceae family, which are aromatic shrubs that reach heights of 2–4 m. The branches of its inflorescence and the calyx are covered with stiff hairs. its leaves are opposite and composed of 3–5 leaflets, which are petiolate and vary in shape from lanceolate to ovate or rhomboid-ovate, measuring 3–13 cm long. These leaves are acute or acuminate at the apex and have a shiny surface on the upper surface. The petals are obovate to ovate, and 3–3.5 mm long. The fruit is a drupe that may be oblong-ovoid or ellipsoidal, often with a narrowed, neck-like base, is 6–14 mm long, and is black in color. *Amyris balsamifera* is native to tropical and subtropical regions of Central America and the Caribbean. It can be found in countries such as Mexico, Belize, Guatemala, Honduras, Nicaragua, Costa Rica, Panama, Cuba, Dominican Republic, Haiti, Jamaica, Puerto Rico, and the Virgin Islands. This tropical tree species thrives in warm, humid areas, and is valued for its aromatic wood and essential oil production [132]. The therapeutic effects of *A. balsamifera* are mainly attributed to the combination of sesquiterpene alcohols contained in its essential oil, which is extracted from the leaves and mainly from the bark. Within the mixture of these alcohols are elemol, 10-epi-γ-eudesmol, R-santalol, turmerona javanol, α-eudesmol, β-eudesmol (β-selinenol), bergamotenol, and valerianol, among others, with valerianol being the most prominent [133]. Traditional medicine in the East and West has used the essential oil of this plant, as it has medicinal properties such as anti-inflammatory activity [134,135]. Additionally, there is scientific evidence that supports the repellent effect of *A. balsamifera* by observing that the essential oil of this plant (in concentrations of 10 and 20%) registered a repellency of 100% that lasted for more than five hours after exposing female mosquitoes of *Ae. aegypti* to a wind tunnel [136]. Furthermore, the repellent effect of *A. balsamifera* oil has also been evaluated using the spatial and contact repellency bioassay tests against *Ae. aegypti*, using a quantitative structure–activity relationship (QSAR) model based on classical and quantum molecular descriptors that include relevant physicochemical properties and structural and electronic features, which are important for understanding the interactions between the compound and the receptor and which allowed the analysis and prediction of the repellent activity. This model identified key properties of the sesquiterpenes present in the oil that influence its effectiveness as a repellent against mosquitoes [137]. Another study also showed that the essential oil of *A. balsamifera* is highly effective as a repellent and exhibits larvicidal activity against *Cx. pipiens* mosquitoes [138]. 

### 2.13. Thyme (Thymus vulgaris)

*Thymus vulgaris*, known as wild thyme or thyme, is an aromatic plant of the genus *Thymus*. It is distinguished by its small green leaves and pink or purple flowers that grow in clusters. This plant is appreciated for its essential oil, which gives it antimicrobial and antioxidant properties. Traditionally, *T. vulgaris* has been used in herbal medicine and in the food industry due to its medicinal properties and its ability to preserve food. *Thymus vulgaris*, or wild thyme, is native to Europe, and can be found in various regions of this continent, including countries such as Spain, France, Italy, Germany, and the United Kingdom, among others. This plant was also introduced to other parts of the world, such as North America, including Mexico. *Thymus vulgaris* prefers to grow in subtropical climates with dry, sunny soils, and is commonly found in meadows, hillsides, and also in open forests [139,140]. *Thymus vulgaris* is a plant with a wide range of medicinal properties, including antimicrobial, antifungal [141], antiparasitic [142], antiviral [143] and antibacterial properties [144]. On the other hand, an analysis of the essential oil of *T. vulgaris* obtained from the leaves and flowers of the plant showed that the main components that constitute the oil are thymol, carvacrol, and cymol, in addition to other non-volatile components such as flavonoid glycosides, caffeic acid oligomers, simple phenolic acids, hydroquinone derivatives, and terpenoids. These components may be responsible for the repellent efficacy against *Ae. aegypti* and *An. stephensi*, since using an olfactometry chamber in which the response of mosquitoes to stimuli from different sources was evaluated, it was observed that the stimuli of mosquitoes towards the essential oil of *T. vulgaris* showed a clear aversion, suggesting a repellent effect [142,145]. However, *T. vulgaris* not only has a repellent effect, but also has a larvicidal effect; this observation was made by using a nanoemulsion of the essential oil of this plant and its encapsulation in chitosan, which was analyzed against third-stage larvae of *Ae. aegypti*, *An. Stephensi*, and *Culex tritaeniorhynchus*, with a high percentage of mortality observed after 24 h of exposure, an effect that lasted up to 3 months with the same efficacy [146].

### 2.14. Wormwood (Artemisia absinthium)

*Artemisia absinthium*, commonly known as wormwood, is a perennial herbaceous plant in the Asteraceae family, characterized by its gray-green leaves and yellow flowers. Native to dry and sunny regions across Europe, Asia, North Africa, North America, South America, and Mexico, it thrives in temperate and subtropical climates with dry, sunny soils [147]. Medicinally, the leaves and stems of *A. absinthium* are used for infusions, aqueous or ethanolic extractions, and essential oil preparations. The essential oil contains a variety of compounds, including monoterpenes, sesquiterpenes, and terpenoids such as thujone, borneol, camphene, z-isocitral, α-pinene, β-pinene, (3E)-octen-2-ol, α-phellandrene, and sabinene, among others, which contribute to its broad range of medicinal properties [148,149]. *Artemisia absinthium* exhibits significant larvicidal activity against various mosquito species, including *An. stephensi*, *An. subpictus*, *Ae. aegypti*, *Ae. albopictus*, *Cx. quinquefasciatus*, and *Cx. tritaeniorhynchus*. Studies have shown high mortality rates in mosquito larvae exposed to the essential oil of this plant and its components, such as (E)-β-farnesene and (Z)-β-ocimene [44]. Additionally, ketone extracts have demonstrated larvicidal, pupicidal, and adulticidal effects against *Ae. aegypti*, disrupting their metamorphic processes [150]. Furthermore, *A. vulgaris*, a close relative, has shown repellent effects against *Ae. aegypti*, with an effectiveness of 63.3% in reducing mosquito bites [151].

### 2.15. Jasmine (Jasminum grandiflorum)

*Jasminum grandiflorum*, commonly known as the grandiflora jasmine, is a climbing plant in the Oleaceae family, reaching heights of 6–7 m with support. Native to Asia and thriving in tropical to subtropical climates, it is cultivated in countries including India, Egypt, Sri Lanka, Myanmar, Morocco, and, to a lesser extent, in France, South Africa, Italy, Spain, Algeria, Turkey, China, and Mexico [152]. The plant’s green, pinnate leaves and fragrant white to pink flowers are utilized for obtaining essential oil, infusions, or methanolic extractions. The essential oil is rich in components such as 9,12-octadecadienoic acid, farnesol, phytol, and linalool, contributing to its medicinal properties, which include anti-inflammatory, antioxidant, healing, and antiulcer effects [153,154]. Additionally, *J. grandiflorum* exhibits a repellent effect against *Ae. aegypti* [130], and its essential oil has demonstrated larvicidal and pupicidal properties, inducing 50% and 90% mortality in mosquito larvae and pupae, respectively, within 24 h [155]. 

### 2.16. Lemon Eucalyptus (Corymbia citriodora)

*Corymbia citriodora*, also known as lemon eucalyptus, is a tall tree in the Myrtaceae family that can reach up to 50 m in height. Native to eastern Australia, it has been introduced to tropical and subtropical regions worldwide, including parts of Africa, Asia, South America, and Mexico [156]. This species is renowned for its medicinal properties, including antimicrobial, anti-inflammatory, analgesic, and relaxing effects [156]. Its essential oil, derived from the leaves and bark, contains monoterpenes such as citronellol and geraniol, which contribute to its anti-inflammatory and analgesic effects. The oil is particularly notable for its mosquito-repellent properties due to including para-menthane-3,8-diol (PMD), a compound with an efficacy comparable to DEET. PMD’s low volatility ensures prolonged protection, making it effective against various mosquito species, including *Aedes*, *Anopheles*, and *Culex*. Bioassays have demonstrated that PMD can provide protection for up to 8 h, and *C. citriodora* essential oil has shown 100% repellent efficacy for 11–12 h against *Ae. aegypti*, *Cx. quinquefasciatus*, and *An. dirus* when applied at a concentration of 1.7 mg/cm^2^ of skin [157,158,159].

## 3. Insecticidal and/or Repellent Effect of Plant-Based Compounds and Their Mode of Action

Plant-derived compounds exhibit a range of effects on mosquitoes, including larvicidal, pupicidal, ovicidal, and repellent properties. These compounds impact various physiological systems in mosquitoes, such as the nervous, respiratory, and endocrine systems, and their water balance. The primary focus of research has been on ovicidal and larvicidal actions, as targeting mosquitoes during their early developmental stages is more effective for eradication before they reach adulthood [160]. Insecticides are typically classified into three categories based on their mode of action [161]:Neurotoxins: These interfere with normal nerve function;Insect Growth Regulators (IGRs): These affect specific physiological systems, hindering the growth and development of insects;Behavior Modifiers: These alter the daily activities and behaviors of insects.

Neurotoxins commonly cause elevated activity levels and excitability in insects, leading to rapid deterioration and eventual immobilization or paralysis [162,163]. These effects are often mediated through mechanisms such as the suppression of chloride channels stimulated by acetylcholinesterase (AChE) and gamma-aminobutyric acid (GABA), the disruption of sodium–potassium ion exchange in nerve cell membranes, the inhibition of calcium channels, and the stimulation of nicotinic acetylcholine and octopamine receptors. This results in a complex interaction of molecular events that disrupt neuronal signaling pathways [27]. 

Plant compounds that act as inhibitors and regulators of growth interfere with the metamorphosis process in insects. Behavior-modifying compounds, including those with repellent properties, impact olfactory neurons and alter host-seeking behaviors when applied to the skin [164]. Approximately 90% of essential oils are composed of monoterpenes, which are identified as the active components responsible for the larvicidal effects of plant-derived substances. These monoterpenes inhibit AChE activity in insects, highlighting their potential in developing new insecticidal agents from natural sources [165].

Despite these insights, research on the modes of action of plant biomolecules in adult mosquito populations remains limited. Future studies should focus on elucidating these mechanisms to enhance the effectiveness of plant-based insecticides. The chemical constituents identified in the plants described, along with their possible modes of action, are summarized in Table 1.

## 4. The Role for Botanical-Based Mosquito Repellent and Control Products in Mexico

Mexico’s rich biodiversity offers a promising foundation for the development of botanical-based mosquito repellents and control products. These natural solutions are likely to be culturally accepted, given the long-standing tradition of using plants for health and wellness in many Mexican municipalities. Plants like citronella, neem, and basil have shown promise as effective tools against mosquito vectors, particularly *Aedes aegypti*, which is responsible for transmitting diseases such as dengue, zika, and chikungunya. However, the question remains: what is the best way to integrate these natural products into mosquito control strategies in Mexico?

One strategy focuses on the cultural acceptance and practical application of these products within local communities. In many Mexican regions, traditional practices already favor natural solutions. Spatial repellents, insecticide-treated nets with plant oils, and environmental treatments like larvicides derived from plants can align well with these cultural preferences [83,157]. This could be combined with the integration of these botanical products into the broader public health system, which still poses challenges. While their natural origins and cultural resonance are advantageous, ensuring their consistent efficacy and safety is crucial. There is a need to rigorously evaluate these products to confirm their effectiveness in real-world conditions and to assess any potential toxicological risks, particularly concerning skin irritation or allergic reactions. This is essential for their acceptance and widespread use within public health programs.

Addressing these concerns also opens the door to innovation in product formulation. Research and development efforts should focus on improving the stability and effectiveness of these products, potentially extending the durations of protection of botanical repellents through advanced formulations such as converting essential oils into creams, lotions, or aerosols, which can significantly improve their practicality [34,136]. However, these advancements come with their own set of challenges, including the potential increase in production costs compared to these products’ synthetic counterparts Ensuring that these products are affordable for populations is essential for their widespread adoption and mitigating the environmental impact of sourcing large quantities of plant materials. In addition to the cost, the environmental impact of harvesting plants for botanical repellents must be considered. Although these products are generally more sustainable than synthetic chemicals, intensive harvesting could harm local ecosystems. It is important to promote sustainable harvesting practices and, where possible, encourage the cultivation of plants used in these repellents. 

The success of plant-based mosquito control in Mexico will depend on a coordinated effort. Collaboration between research institutions, government agencies, and the private sector is crucial to drive both the scientific development and the practical implementation of these products. By addressing issues of accessibility, regulation, and sustainability, Mexico can effectively leverage its natural resources to combat mosquito-borne diseases in a sustainable and culturally sensitive manner.

## 5. Conclusions

This review significantly contributes to expanding our knowledge of plant-derived bioinsecticides and repellents, paving the way for exploring next-generation strategies for controlling mosquito vectors of disease. These strategies aim at more effective and sustainable pest-management practices.

The findings are promising, suggesting the need for further investigation into various botanical specimens and the validation of their results in animal models and human applications. Future studies must delve into the pharmacological profiles of plants to understand their repellent and insecticidal properties, identifying key molecules for different formulations and establishing potential mechanisms of action. This will promote their application in health products with therapeutic activity, positively impacting the interruption of mosquito-borne disease transmission. Emphasizing the crucial role mosquitoes play in spreading these arboviruses highlights the beneficial implications for public health.

Plant compounds can serve as suitable alternatives to synthetic repellents and insecticides in the future, being relatively safe and readily available in Mexico and other parts of the world. However, it is fundamental to acknowledge the long research path ahead. While there are promising starts towards reducing environmental pollution and the sustainable use of bioactive plant compounds from Mexican-cultivated plants, interdisciplinary and multidisciplinary approaches are required for the exponential development of biodirected research, enabling the identification of active compounds, pharmacodynamics, and toxicological studies for repellents, as well as research aimed at accelerating the industrialization and production of these compounds. Current advancements do not yet allow for the implementation of these compounds for community use.

## Figures and Tables

**Figure 1 pathogens-13-00737-f001:**
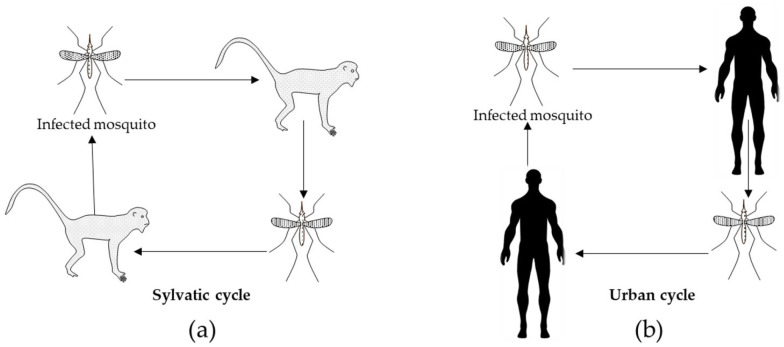
Transmission cycles of arboviruses: dengue, zika, and chikungunya. (**a**) Sylvatic transmission cycle, the virus is transmitted between the vector and animals. (**b**) Urban cycle, the virus is transmitted between the vector and humans.

**Table 1 pathogens-13-00737-t001:** Identification of metabolites and components of the plant used, and their possible mode of action.

Species	Used Part of the Plant	Effect	Type of Extract	Biomarker	Suggested Mode of Action	Reference
*Citronella (Cymbopogoncu nardus)*	Leaves, roots, aerial parts and rhizomes	Repellent and insecticide	Essential oil	Monoterpenes *, monoterpenoids *, sesquiterpenes **, sesquiterpenoids and some fatty alcohols such as 1-octanol and 4-nonanol.	Neurotoxic * and Behavior Modifiers **	[40,41,81,160]
*Pepper (Piper aduncum)*	Leaves and stems	Repellent and insecticide (larvicide, adulticide)	Essential oil	Dillapiol *, γ-Terpinene, p-Cymene, Limonene, α-Pinene and β-Pinene *, (E)-β-caryophyllene	Neurotoxic * and Behavior Modifiers	[50,81,160,166]
*Black Pepper (Piper nigrum)*	Seeds	Repellent and insecticide (larvicide)	Essential oil	(E)-caryophyllene, limonene, sabinene, α-terpineol **, borneol and terpinen-4-ol, β-caryophyllene, α-pinene **, β-pinene **, 3-carene, eugenol ** andmethyl eugenol *	Neurotoxic * and Behavior Modifiers **	[55,81,160]
*Garlic (Allium sativum)*	Bulb segments	Repellent and insecticide (larvicide)	Extracts and essential oils	dimethyl trisulfide, diallyl disulfide *, diallyl sulfide *, diallyl tetrasulfide and 3-vinyl-[4H]-1,2-dithiine	Neurotoxic * and Behavior Modifiers	[61]
*Lemongrass (Cymbopogon citratus)*	Leaves or the entire plant	Repellent and insecticide (larvicide)	Essential oil	Citral α, Citral β, Nerol, Geraniol *, Citronellal *, Terpinolene, geranyl acetate, Mirecene, Terpinol **, Methylheptenone, limonene	Neurotoxic * and Behavior Modifiers **	[72,73,81,160,167]
*Cinnamon (Cinnamomum zeylanicum)*	Bark, leaves and root bark	Repellent	Essential oil	Transcinnamaldehyde, theeugenol **, and linalool *	Neurotoxic * and Behavior Modifiers **	[88]
*Neem (Azadirachta indica)*	Seeds	Repellent and insecticide	Essential oil	Azadirachtin ***, meliantriol, salanin, desacetyl salanin, nimbin, desacetyl nimbin, nimbidine, and nimbolides	IGR ***	[81,100,160]
*Basil (Ocimum basilicum)*	Leaves and stems	Repellent and insecticide (larvicide)	Extracts and essential oils	Linalool *, thymol *, cineole * or eucalyptol, methyleugenol and eugenol **, methylchavicol, 2-(2-butoxyethoxy) ethanol, cedrelanol, 2,4, di-tert-butylphenol and phytol	Neurotoxic * and Behavior Modifiers **	[81,105,106,160]
*Guava (Psidium guajava)*	Leaves	Insecticide (larvicide)	Extracts	β-caryophyllene	Neurotoxic, IGR or Behavior Modifier	[168,169]
*Peppermint (Mentha piperita)*	Leaves	Repellent and insecticide (larvicide, adulticide)	Extracts and essential oils	Menthol, menthone, neomenthol and iso-menthone	Neurotoxic, IGR or Behavior Modifier	[124,125,170]
*Chamomile (Anthemis nobilis)*	Flowers, leaves and stems	Repellent and insecticide (adulticide)	Essential oil	α-bisabolol, α-pinene ** and β-pinene **, chamazulene, en-yne-dicycloether, farnesene, spiroether and various esters	Behavior modifiers **	[81,129,160]
*Jamaican Rosewood (Amyris balsamifera)*	Leaves and bark	Repellent and insecticide (larvicide)	Essential oil	Elemol, 10-epi-γ-eudesmol, R-santalol, Turmerone javanol, α-eudesmol, β-eudesmol (β-selinenol), Bergamotenol, Valerianol	Neurotoxic, IGR or Behavior Modifier	[81,133,160]
*Thyme (Thymus vulgaris)*	Leaves and flowers	Repellent and insecticide (larvicide)	Essential oil	Monoterpenes *, thymol *, carvacrol, and cymene	Neurotoxic * and Behavior Modifiers	[81,142,160,171]
*Wormwood (Artemisia absinthium)*	Leaves and stems	Repellent and insecticide (larvicide, pupicide and adulticide)	Extracts and essential oils	Monoterpenes *, sesquiterpenes **, terpenoids (thujone, borneol, cinelol, camphene, z-isocitral, α-pinene **, β-pinene ** (3E)-octen-2-ol, α-phellandrene, sabinene)	Neurotoxic * and Behavior Modifiers **	[81,149,160,172]
*Jasmine (Jasminum grandiflorum)*	Leaves and flowers	Repellent and insecticide (larvicide, pupicide)	Extracts and essential oils	9,12-octadecadienoic acid (Z,Z), 11,14,17-eicosatrienoic acid, farnesol isomer, phytol **, stigmast-5-en-3-ol, trimethoxy benzaldehyde, Cis-9,12,15-octadecatrienoic (linolenic acid), α-hexylcinnamaldehyde, nerolidol, hexahydrofarnesol acetone, benzyl acetate, benzyl benzoate and linalool *	Neurotoxic * and Behavior Modifiers **	[81,153,154,160]
*Lemon eucalyptus (Corymbia citriodora)*	Leaves and bark	Repellent	Essential oil	Citronellol, geraniol, para-menthane-3,8-diol (PMD) **	Behavior modifiers **	[157]

* Molecule responsible for the insecticidal effect; ** molecule responsible for the repellent effect; *** IGR: insect growth regulators.

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
