# Peer review of "A Review of Botanical Extracts with Repellent and Insecticidal Activity and Their Suitability for Managing Mosquito-Borne Disease Risk in Mexico"

_pathogens, 2024, doi:10.3390/pathogens13090737_

Round 1

Reviewer 1 Report

Comments and Suggestions for Authors

Dear Authors, 

The submitted manuscript is well-written in good and understandable English. It represents an important database about the insecticidal and repellent effects of botanical products. I enjoyed reading it. 

Before it is published, this material should be revised to eliminate technical mistakes. Please find below my specific comments. 

When insect species is mentioned for the first time in text, it should be given author and year when it was described.

L27 It is more appropriate to say – by arthropods instead only ticks and mosquitoes, but authors can give these two in the bracket as an example.

L38 Reported where?

L52 Please correct Ttoday

L88 Missing citation for - infection rate of approximately 1 per 1000 eggs.

Figure 1. Transmission cycles of arboviruses.- This is too general. Many arboviruses are not transmitted that way. Please be specific. Also in the text when you talk about cycles. Generalizing is not correct.

L100 and L101 Please give full name at the beginning of the sentence.

L100 and L101 Missing space after full stop. Example Ae.albopictus. Same for lines 227

L114-116  Ae. albopictus has diurnal biting activity. Please correct this.

L117 ...that transmit what? (1.3 Strategies for the prevention and control of mosquitoes that transmit)

L123 Fumigation for mosquitoes is most common? Please revise this.

L126 Please correct to ...of water bodies

L127 Did you mean pollution instead of composition? If not, the context is not clear.

L139 It is not Physical barriers but mechanical barriers

L203 Please give the family that selected plants belong to.

L230 Please correct Ae. Aegypti.

L259 Please remove comma after square bracket and delete full stop.

L267 Please correct Ae. Aegypti

L296 Please replace as; with as:

L302 It didn’t work investigated, but the Samuel et al. did.

L308-317 This is too long sentence and the last part of it is not clear (information that makes sense with the results reported by Tawatsin et al., regarding its slight protective effect against the bite of the Ae. aegypti mosquito in particular [50]). Please be specific in the last part of the sentence.

L314 Culex is abbreviated Cx. and not C. Please correct accordingly in the whole manuscript.

L319 Please correct Allium Sativum

L323 Sentence cannot start with the abbreviated name. Please correct accordingly in the whole manuscript. Same in L360

L326 Space is missing between the numbers in bracket.

L349 Please correct An. Stephensi

L348 Unit is missing. Probably ppm?

L369 Please correct- properties. [67],

L369 Missing comma after brackets.

L372 Delete comma- Kimutai et al.,

L377 Missing author and year after species name

L378 In italics - C. citrates

L383 -  2.6. Please delete full stop in bracket

L418 Aedes in italics

L421  Please correct - Ae. Aegypty to correct name and small letters.

L448 Missing space- Ae.aegypti

L450 Please correct- (Ocimum Basilicum)

L473 Please correct- Ae. Aegypti

L487 Please correct-  [101]. furthermore

L491 Please delete “it”

L505 Please modify - Due to its widely used medicinal use....

L514 Please delete commas Rwang et al.,.... Please do the same in the whole manuscript.

L520 Please correct Ae. Aegypti

L526 Abbreviation at the beginning. Please correct. Same in L531.

L553 Please abbreviate Culex quinquefasciatus.

L555 Please correct (Anthemis Nobilis)

L576 Anastrepha suspensa is not mosquito. It is Caribbean fruit fly.

L578 Please abbreviate it - Anastrepha suspense

L579 Correct- (Amyris Balsamifera)

L612 Culex pipiens should be abbreviated

L617 T. vulgaris in italics

L646 Delete the colour.

L654-656 Some species should be abbreviates. Missing space after full stop.

L686 Please correct (Jasminum Grandiflorum)

L720 Shorten the name of Culex species.  

L722 Make in superscript letters 2 in cm2 in the whole manuscript.

L734-735 What is this o?

Table 1. Please correct capitalize letters of the species.

References

29. All letters are capitalized.

31. This reference is about Aurantifolia and not about Cymbopogoncus nardus. Please replace or just delete.

Comments on the Quality of English Language

English is ok. 

Author Response

Reviewer 1:

  1. When insect species is mentioned for the first time in text, it should be given author and year when it was described.

Our response: Thank you for your observation. We have added that information.

  1. L27 It is more appropriate to say – by arthropods instead only ticks and mosquitoes, but authors can give these two in the bracket as an example.

Our response: Thank you for your comment. We have modified.

  1. L38 Reported where?

Our response: We have added that particular piece of data.

  1. L52 Please correct Ttoday

Our response: We have addressed the observation.

  1. L88 Missing citation for - infection rate of approximately 1 per 1000 eggs.

Our Response: Thank you for your observation. We have added the references to the paragraph.

  1. Figure 1. Transmission cycles of arboviruses.- This is too general. Many arboviruses are not transmitted that way. Please be specific. Also in the text when you talk about cycles. Generalizing is not correct.

Our response: Thank you for your comment. We have expanded the information on transmission specifically for DENV, ZIKV and CHIKV.

  1. L100 and L101 Please give full name at the beginning of the sentence.

Our response: Thank you for your observation We have corrected the sentence.

  1. L100 and L101 Missing space after full stop. Example Ae.albopictus. Same for lines 227

Our response: We have corrected the error.

  1. L114-116   albopictus has diurnal biting activity. Please correct this.

Our response: Thank you for your comment. We have incorporated this into the information

  1. L117 ...that transmit what? (1.3 Strategies for the prevention and control of mosquitoes that transmit)

Our response: Thank you for your observation, we have supplemented the information.

  1. L123 Fumigation for mosquitoes is most common? Please revise this.

Our response: Thank you for your contribution, we have modified the text and added more information about it.

  1. L126 Please correct to ...of water bodies

Our response: Thank you, we have corrected it.

  1. L127 Did you mean pollution instead of composition? If not, the context is not clear.

Our response: Thank you, we have corrected the wording of the paragraph.

  1. L139 It is not Physical barriers but mechanical barriers

Our response: Thank you for your comment, we have modified it.

  1. L203 Please give the family that selected plants belong to.

Our response: We have added the information.

  1. L230 Please correct Ae. Aegypti.

Our response: We have corrected it.

  1. L259 Please remove comma after square bracket and delete full stop.

Our response: We have removed it.

  1. L267 Please correct Ae. Aegypti

Our response: We have corrected the error.

  1. L296 Please replace as; with as:

Our response: We have replaced it.

  1. L302 It didn’t work investigated, but the Samuel et al. did.

Our response: Thank you for your comment. The wording of the paragraph was modified.

  1. L308-317 This is too long sentence and the last part of it is not clear (information that makes sense with the results reported by Tawatsin et al., regarding its slight protective effect against the bite of the Ae. aegypti mosquito in particular [50]). Please be specific in the last part of the sentence.

Our response: Thank you for your observation. The information was shortened and the wording of the paragraph was modified.

  1. L314 Culex is abbreviated Cx. and not C. Please correct accordingly in the whole manuscript.

Our response: We have corrected the error in the manuscript.

  1. L319 Please correct Allium Sativum

Our response: We have corrected it.

  1. L323 Sentence cannot start with the abbreviated name. Please correct accordingly in the whole manuscript. Same in L360

Our response: We have corrected it. We have corrected, starting with the full name in the whole manuscript.

  1. L326 Space is missing between the numbers in bracket.

Our response: We have corrected the error.

  1. L349 Please correct An. Stephensi

Our response: We have corrected it.

  1. L348 Unit is missing. Probably ppm?

Our response: Thank you for your observation. We have added the units.

  1. L369 Please correct- properties. [67],

Our response: We have corrected the writing.

  1. L369 Missing comma after brackets.

Our response: We have added the comma after the brackets.

  1. L372 Delete comma- Kimutai et al.,

Our response: We have deleted the comma.

  1. L377 Missing author and year after species name

Our response: Thank you for your observation. We have added that information.

  1. L378 In italics - C. citrates

Our response: We have made the modification in italics.

  1. L383 -  2.6. Please delete full stop in bracket

Our response: We have removed the full stop in brackets.

  1. L418 Aedes in italics

Our response: We have made the modification in italics.

  1. L421  Please correct - Ae. Aegypty to correct name and small letters.

Our response: We have corrected the name.

  1. L448 Missing space- Ae.aegypti

Our response: We have added the space.

  1. L450 Please correct- (Ocimum Basilicum)

Our response: We have corrected it.

  1. L473 Please correct- Ae. Aegypti

Our response: We have corrected it.

  1. L487 Please correct-  [101]. Furthermore

Our response: We have corrected it with capital letters.

  1. L491 Please delete “it”

Our response: Thanks for the comment, it has been deleted.

  1. L505 Please modify - Due to its widely used medicinal use....

Our response: Thank you for the observation, the text has been modified.

  1. L514 Please delete commas Rwang et al.,.... Please do the same in the whole manuscript.

Our response: We have removed commas after authors throughout the manuscript.

  1. L520 Please correct Ae. Aegypti

Our response: We have corrected it.

  1. L526 Abbreviation at the beginning. Please correct. Same in L531.

Our response: We have corrected it. We have corrected, starting with the full name in the whole manuscript.

  1. L553 Please abbreviate Culex quinquefasciatus.

Our response: the genus was abbreviated.

  1. L555 Please correct (Anthemis Nobilis)

Our response: We have corrected it.

  1. L576 Anastrepha suspensa is not mosquito. It is Caribbean fruit fly.

Our response: We have modified the writing considering your observation, thank you.

  1. L578 Please abbreviate it - Anastrepha suspense

Our response: we have modified with the abbreviation.

  1. L579 Correct- (Amyris Balsamifera)

Our response: We have corrected it.

  1. L612 Culex pipiens should be abbreviated

Our response: we have modified with the abbreviation.

  1. L617 T. vulgaris in italics

Our response: We have added in italics.

  1. L646 Delete the colour.

Our response: The color that appeared has been removed, thank you.

  1. L654-656 Some species should be abbreviates. Missing space after full stop.

Our response: The species were abbreviated and a space was placed after full stop.

  1. L686 Please correct (Jasminum Grandiflorum)

Our response: We have corrected it.

  1. L720 Shorten the name of Culex species.  

Our response: We have corrected it.

  1. L722 Make in superscript letters 2 in cm2 in the whole manuscript.

Our response: Thank you for your comment, the observation has been modified throughout the manuscript.

  1. L734-735 What is this o?

Our response: They were meant to be cartoons, but they have already been modified in the text.

  1. Table 1. Please correct capitalize letters of the species.

Our response: We have corrected it.

  1. Reference 29. All letters are capitalized.

Our response: Thank you for your comment, we have modified it.

  1. Reference 31. This reference is about Aurantifolia and not about Cymbopogoncus nardus. Please replace or just delete.

Our response: We have added the corresponding reference, thank you for your comment.

Reviewer 2:

  1. Apart from some brief mentions of Mexico and the rates of mosquito-borne disease, I don't think the paper adequately addresses the situation in Mexico and I don't think it is provided sufficient discussion in text given the title states it will provide "a comprehensive study of arboviruses in Mexico". There is much more to discuss and a seperate paper could be developed that delves further into the environmental and social factors that drive activity of arboviruses in that country. As it stands, the material included here does service as a good introduction to the situatioin in mexico and some background as to why plant-based repellents may need to be considered.

Our response: Indeed, this observation is correct, and the title of this work has been changed to " A comprehensive analyzing plants with repellent and insecticidal activity with potential to combat arboviruses in Mexico". Now, as can be seen, the title is in line with the main objective, and both sections are more in line with the theoretical depth that underlies why it is necessary to consider plant-based repellents. We appreciate and value this observation that helped to improve the understanding of the scope of this work, since the primary objective of this work was not to address the situation of arboviruses in Mexico in the first instance.

  1. The bulk of the paper does address the various botanical-based products that have been used as agents to reduce mosquito bites. There is, however, a substantial amount of information available already on these products. I think ti would have been far more important to include a detailed discussion about how these products can be locally significant in mosquito management programs. Would the community use topical or non-topical formulations of these products? Could local authorities switch from commonly used insecticides to these botanical based products? What would the process be to confirm efficacy of these products in the local setting? 

Our response: In consideration of your valuable feedback, we have added responses to your questions and observations in the perspectives and conclusions section. Thank you.

Minor comments:

  1. Title: Change “Arbovirosis” to “Arboviruses”

Our response: We have modified the title.

  1. Line 38: There is variability in the capitalisation of “zika”, check manuscript for consistency.

Our response: We have modified throughout the manuscript.

  1. Line 50: Change “february” to “February”

Our response: We have corrected it.

  1. Line 52: Change “Ttoday” to “Today”

Our response: We have corrected the error.

  1. Line 227: Should “An. quinquefasciatus” be “Cx.quinquefasciatus”?

Our response: We have modified and corrected the error.

  1. Line 276: Change “Ae. Aegypti” to” Ae. aegypti” (note that there is a number of other locations in manuscript where similar formatting errors with scientific names and capitalisation has occurred. There is a need to carefully review manuscript. This also includes a need to ensure all appropriate abbreviations of genus names is included (e.g. Culex – Cx.)

Our response: We have modified throughout the manuscript.

Reviewer 2 Report

Comments and Suggestions for Authors

I appreciate that considerable work has gone into the review of these plant-based substances and their potential for reducing mosquito bites, either through insecticidal activity or repellency. However, there is already a substantial amount of scientific literature outlining these products and their relative importance in mosquito bite prevention programs.

Apart from some brief mentions of Mexico and the rates of mosquito-borne disease, I don't think the paper adequately addresses the situation in Mexico and I don't think it is provided sufficient discussion in text given the title states it will provide "a comprehensive study of arboviruses in Mexico". There is much more to discuss and a seperate paper could be developed that delves further into the environmental and social factors that drive activity of arboviruses in that country.

As it stands, the material included here does service as a good introduction to the situatioin in mexico and some background as to why plant-based repellents may need to be considered.

The bulk of the paper does address the various botanical-based products that have been used as agents to reduce mosquito bites. There is, however, a substantial amount of information available already on these products. I think ti would have been far more important to include a detailed discussion about how these products can be locally significant in mosquito management programs. Would the community use topical or non-topical formulations of these products? Could local authorities switch from commonly used insecticides to these botanical based products? What would the process be to confirm efficacy of these products in the local setting? 

Minor comments:

Title: Change “Arbovirosis” to “Arboviruses”

Line 38: There is variability in the capitalisation of “zika”, check manuscript for consistency.

Line 50: Change “february” to “February”

Line 52: Change “Ttoday” to “Today”

Line 227: Should “An. quinquefasciatus” be “Cx.quinquefasciatus”?

Line 276: Change “Ae. Aegypti” to” Ae. aegypti” (note that there is a number of other locations in manuscript where similar formatting errors with scientific names and capitalisation has occurred. There is a need to carefully review manuscript. This also includes a need to ensure all appropriate abbreviations of genus names is included (e.g. Culex – Cx.)

Comments on the Quality of English Language

no specific comment.

Author Response

Reviewer 2:

  1. Apart from some brief mentions of Mexico and the rates of mosquito-borne disease, I don't think the paper adequately addresses the situation in Mexico and I don't think it is provided sufficient discussion in text given the title states it will provide "a comprehensive study of arboviruses in Mexico". There is much more to discuss and a seperate paper could be developed that delves further into the environmental and social factors that drive activity of arboviruses in that country. As it stands, the material included here does service as a good introduction to the situatioin in mexico and some background as to why plant-based repellents may need to be considered.

Our response: Indeed, this observation is correct, and the title of this work has been changed to " A comprehensive analyzing plants with repellent and insecticidal activity with potential to combat arboviruses in Mexico". Now, as can be seen, the title is in line with the main objective, and both sections are more in line with the theoretical depth that underlies why it is necessary to consider plant-based repellents. We appreciate and value this observation that helped to improve the understanding of the scope of this work, since the primary objective of this work was not to address the situation of arboviruses in Mexico in the first instance.

  1. The bulk of the paper does address the various botanical-based products that have been used as agents to reduce mosquito bites. There is, however, a substantial amount of information available already on these products. I think ti would have been far more important to include a detailed discussion about how these products can be locally significant in mosquito management programs. Would the community use topical or non-topical formulations of these products? Could local authorities switch from commonly used insecticides to these botanical based products? What would the process be to confirm efficacy of these products in the local setting? 

Our response: In consideration of your valuable feedback, we have added responses to your questions and observations in the perspectives and conclusions section. Thank you.

Minor comments:

  1. Title: Change “Arbovirosis” to “Arboviruses”

Our response: We have modified the title.

  1. Line 38: There is variability in the capitalisation of “zika”, check manuscript for consistency.

Our response: We have modified throughout the manuscript.

  1. Line 50: Change “february” to “February”

Our response: We have corrected it.

  1. Line 52: Change “Ttoday” to “Today”

Our response: We have corrected the error.

  1. Line 227: Should “An. quinquefasciatus” be “Cx.quinquefasciatus”?

Our response: We have modified and corrected the error.

  1. Line 276: Change “Ae. Aegypti” to” Ae. aegypti” (note that there is a number of other locations in manuscript where similar formatting errors with scientific names and capitalisation has occurred. There is a need to carefully review manuscript. This also includes a need to ensure all appropriate abbreviations of genus names is included (e.g. Culex – Cx.)

Our response: We have modified throughout the manuscript.

Round 2

Reviewer 2 Report

Comments and Suggestions for Authors

I have reviewed the revised manuscript titled "A comprehensive analyzing plants with repellent and insecticidal activity with potential to combat  arboviruses in Mexico". As I stated in an earlier review of this manuscript, this paper does not present a thorough presentation is issues associated with mosquito-borne disease in Mexico, nor a clear indication of why this review of botanical-based repellents and insecticides is of practical relevance to Mexico. I do think that there is valuable information contained within the manuscript but there remains some substantial review and revision to improve overall quality.

I think that it is important to note that there is already a substantial amount of literature available on plant-based mosquito control products BUT there is a paucity of information on these products' use in mexico. As a consequence, greater focus needs to be paid to the local context.

I'm providing the following recommendations.

1. I suggest changing to the title of this manuscript to "A review of botanical extracts with repellent and insecticidal activity and their suitability for managing mosquito-borne disease risk in Mexico"

2. To assist in creating a more concise article, the sections titled "1.1 Transmission of arboviruses: dengue, zika and chikungunya" and "1.2 Origin and expansion of mosquito vectors" can be removed from manuscript. This information is available elsewhere and is not necessarily beneficial to this manuscript and its topic.

3. The section titled "1.3 Strategies for the prevention and control of mosquitoes that transmit arboviruses" should be revised to include very specific comments on the way that mosquito-borne disease is managed in Mexico. This can include information provided on surveillance and control programs. While not necessarily comprehensive, the details currently provided on various management strategies and approaches should be placed within the context of Mexico.

4. An additional section should be added titled "The role for botanical-based mosquito repellent and control products in Mexico". The material already added to the "Perspectives and Conclusions" section can be moved into this new section. However, rather than simply listing the issues that need to be addressed, some discussion on how best to address these issues should be added. 

As previously stated, there needs to be a thorough review of the manuscript, despite stating that they had addressed this issue in the revised manuscript, there are still many formatting errors of scientific names.

I have not listed specific minor corrections required. However, it is important to note that in line 598, "Ixodes" is listed - this is not a genus of mosquitoes but, rather, a genus of ticks. Text should be modified accordingly. 

Comments on the Quality of English Language

No specific comment. Manuscript should be reviewed for grammatical, typographical, and presentation issues. 

Author Response

We appreciate the comments, however, it would have been great if these had been considered from the first review. In response to them, here are my comments:

  1. I suggest changing to the title of this manuscript to "A review of botanical extracts with repellent and insecticidal activity and their suitability for managing mosquito-borne disease risk in Mexico"

Our response: Thank you very much for your suggestion, we have modified the title.

  1. To assist in creating a more concise article, the sections titled "1.1 Transmission of arboviruses: dengue, zika and chikungunya" and "1.2 Origin and expansion of mosquito vectors" can be removed from manuscript. This information is available elsewhere and is not necessarily beneficial to this manuscript and its topic.

Our response: Thank you for your comment. But, this information was added because many arboviruses are transmitted in a different way, for this reason we are specific, also to indicate the relevance of Aedes aegypti and its behavior as the responsible vector in the American continent, we consider that both the subtopic 1.1 Transmission of arboviruses: dengue, zika and chikungunya" and "1.2 Origin and expansion of mosquito vectors" are important in the manuscript.

  1. The section titled "1.3 Strategies for the prevention and control of mosquitoes that transmit arboviruses" should be revised to include very specific comments on the way that mosquito-borne disease is managed in Mexico. This can include information provided on surveillance and control programs. While not necessarily comprehensive, the details currently provided on various management strategies and approaches should be placed within the context of Mexico.

Our response: Thank you for your observation. We have added the information.

  1.     An additional section entitled "The role of botanical-based mosquito repellent and control products in Mexico" should be added. The material already added to the "Perspectives and Conclusions" section can be moved to this new section. However, rather than simply listing the issues that need to be addressed, a discussion of how best to address them should be added. Our response: Thank you for your comment. We have added an additional section entitled "The role of botanical-based mosquito repellent and control products in Mexico" As stated above, a thorough review of the manuscript is needed, despite stating that this issue had been addressed in the revised manuscript, there are still many formatting errors in scientific names. Our response: Thank you for your comment, we have reviewed it again. I have not listed the specific minor corrections required. However, it is important to note that on line 598, "Ixodes" is listed, this is not a genus of mosquitoes, but rather a genus of ticks. The text should be amended accordingly. Our response: Thank you for your comment, you are right, these are arthropods, not just mosquitoes. Due to the relevance of the information according to the objective of this review, Ixodes was simply removed from the paragraph. Please see the attached file.

Round 3

Reviewer 2 Report

Comments and Suggestions for Authors

Thank you for addressing comments on previous reports.